# Biodiversity and Ecosystem Function under Simulated Gradient Warming and Grazing

**DOI:** 10.3390/plants11111428

**Published:** 2022-05-27

**Authors:** Zhonghua Zhang, Li Ma, Xiaoyuan Yang, Qian Zhang, Yandi She, Tao Chang, Hongye Su, Jian Sun, Xinqing Shao, Huakun Zhou, Xinquan Zhao

**Affiliations:** 1Northwest Institute of Plateau Biology, Chinese Academy of Sciences, Xining 810001, China; zhangzhonghua@nwipb.cas.cn (Z.Z.); mali@nwipb.cas.cn (L.M.); yangxy96163@163.com (X.Y.); zhangqian@nwipb.cas.cn (Q.Z.); sheyandi@nwipb.cas.cn (Y.S.); changtao@nwipb.cas.cn (T.C.); suhongye@nwipb.cas.cn (H.S.); 2Key Laboratory of the Cold Regions Restoration Ecology, Xining 810001, China; sunjian@itpcas.ac.cn (J.S.); shaoxinqing@163.com (X.S.); 3College of Resources and Environment, University of Chinese Academy of Sciences, Beijing 100049, China; 4Institute of Tibetan Plateau Research, Chinese Academy of Sciences, Beijing 100101, China; 5College of Grassland Science and technology, China Agricultural University, Beijing 100083, China

**Keywords:** alpine meadows, biodiversity, microorganisms, phylogenetic diversity, productivity

## Abstract

Biodiversity and ecosystem functions and their relationship with environmental response constitute a major topic of ecological research. However, the changes in and impact mechanisms of multi-dimensional biodiversity and ecosystem functions in continuously changing environmental gradients and anthropogenic activities remain poorly understood. Here, we analyze the effects of multi-gradient warming and grazing on relationships between the biodiversity of plant and soil microbial with productivity/community stability through a field experiment simulating multi-gradient warming and grazing in alpine grasslands on the Tibetan Plateau. We show the following results: (i) Plant biodiversity, soil microbial diversity and community productivity in alpine grasslands show fluctuating trends with temperature gradients, and a temperature increase below approximately 1 °C is beneficial to alpine grasslands; moderate grazing only increases the fungal diversity of the soil surface layer. (ii) The warming shifted plant biomass underground in alpine grasslands to obtain more water in response to the decrease in soil moisture caused by the temperature rise. Community stability was not affected by warming or grazing. (iii) Community stability was not significantly correlated with productivity, and environmental factors, rather than biodiversity, influenced community stability and productivity.

## 1. Introduction

The most unique feature of the Earth is the existence of life, and the most extraordinary feature of life is its diversity [1]. Biodiversity is fundamental to the survival and development of humanity and reflects many ecological and evolutionary processes occurring at different spatial and temporal scales [2], but the human-induced loss and fragmentation of plant and animal habitats, overexploitation of resources, climate change, and biological invasions have emerged in recent years as important factors in the unprecedented loss of biodiversity and decline in ecosystem functions [3]. In this scenario, ensuring the provision and stability of ecosystem services related to biodiversity (e.g., food, fodder, carbon sequestration and soil fertility) is an urgent socio-ecological issue in the wake of climate change [4].

Extensive preliminary research has shown that biodiversity influences primary productivity and other ecosystem functions [5,6,7,8,9,10]. Tilman et al. [5] showed that species richness had a significant positive effect on plant biomass, as did Hector et al. [6] who showed that the number and type of species (functional groups) present in grassland communities had a relatively similar role in determining aboveground productivity. Furthermore, biodiversity can also make ecosystems more stable [11]. In the 1950s, MacArthur et al. [12] proposed that communities containing more species should be better protected from the effects of species extinction, species invasion, or environmental disturbance. However, most previous studies on the relationship between biodiversity and ecosystem function focused on “richness” (i.e., number of species), and few on “biodiversity” or “diversity” (a measure of the combination of species number and relative abundance) [13]. Moreover, it is well known that ecosystem biodiversity should include the diversity of organisms at multiple trophic levels, and there are several interactions among these biodiversity indexes that affect ecosystem function (productivity) [14,15]. For example, plant biodiversity affects microbial diversity by influencing the state of substrates (soil organic matter) decomposed by microorganisms [16,17], and microorganisms exert feedback on plant community composition by altering soil nutrients [18]. Finally, the interaction of plants and microorganisms can collaboratively influence ecosystem function (vegetation productivity, soil carbon and soil nitrogen stocks, etc.) [19,20,21]. In addition, the importance of biodiversity to ecosystems has recently been questioned, as observational and experimental studies seem to have yielded different results on the effects of biodiversity on ecosystem functioning [22,23]. Some experimental studies have shown that the availability of limited terrestrial resources (e.g., soil nutrients and water) has a greater impact on ecosystem functioning (e.g., community productivity and productivity stability) than species diversity [24,25]. Thus, there are many gaps in the research on the relationship between biodiversity and ecosystem functioning. Ecosystems around the world are exposed to intense warming and human activity (grazing of grasslands, deforestation, expansion of building sites, etc.), but the magnitude of warming and the manner of anthropogenic disturbance vary from region to region. In this scenario, it is particularly important to analyze the processes and mechanisms by which biodiversity affects ecosystem function (productivity and stability) in fluctuating environments [26]. A full understanding of the factors that influence ecosystem function and the importance of biodiversity can ensure the conservation and restoration of diverse natural ecosystems [27].

Nearly half of the Qinghai-Tibet Plateau is covered by grasslands, one of the most important areas for biodiversity research, and organisms in this region are sensitive to environmental disturbances. In recent years, the Qinghai-Tibet Plateau has experienced severe warming and anthropogenic disturbance, and due to the characteristics of seasonal grazing on the Tibetan plateau, anthropogenic grazing disturbance has remained at a relatively strong level in recent years, resulting in the grasslands of the Tibetan Plateau undergoing significant changes. To reveal the relationship between biodiversity and ecosystem function in the alpine grasslands of the Tibetan Plateau in this context, this study analyzes the changes in biodiversity and ecosystem function (productivity and stability) of grasslands through an interactive experiment with simulated gradient warming and simulated grazing in an alpine grassland conducted for eight years in order to answer the following two questions: (1) How do biodiversity and ecosystem function (productivity and stability) change under the influence of warming and simulated grazing? (2) Is there a correlation between biodiversity and ecosystem function and what factors influence both?

## 2. Results

### 2.1. Effect of Simulated Gradient Warming and Grazing on Environmental Factors

Soil organic matter (SOM) is a product of aboveground plants, but it is also a habitat for soil microorganisms and can affect the ground state of the soil [28], which is why it was used as an environmental factor in this study. We used different sizes of Open Top Chamber (OTC) to achieve different temperature increases (Figure 1). The temperature at 5 cm above the soil (T.AG5) differed significantly among treatments (Appendix A), but the temperature showed an anomalous trend of increase and subsequent decrease with the experimental treatments, in contrast to the temperature at 10 cm above the soil (T.AG10); the soil temperature at 5 cm above the ground (T.BG5) showed a gradual increase with the experiment; soil moisture content (SMC) then showed a corresponding gradual decrease; and soil bulk density (SBD) showed an insignificant gradual increase with the warming gradient. Grazing increased the aboveground temperature but reduced soil temperature and soil moisture content. It is noteworthy that the change in soil organic matter content, which showed a U-shaped trend with temperature change when the gradient warmed without grazing, showed a hump-shaped trend after grazing, and grazing completely changed the way SOM responded to gradient warming. Of all the environmental factors, only the temperature at 5 cm above the ground and soil moisture content were significantly affected by gradient warming, and the other environmental factors did not change significantly with gradient warming. Simulated grazing did not significantly change all environmental factors, and the interaction between gradient warming and simulated grazing had no significant effect on environmental factors (Appendix A).

### 2.2. Effects of Simulated Gradient Warming and Grazing on Ecosystem Structure and Function

The Shannon diversity, Simpson diversity, and Pielou diversity indices of plant communities were more consistently affected by the gradient warming and simulated grazing (Figure 2); all showed a fluctuating trend of decreasing then increasing then decreasing with temperature; the maximum warming had the least diversity, but only the Shannon diversity index and Simpson diversity showed significant changes under different gradients, and grazing also had a non-significant negative effect on plant diversity (Appendix A).

The phylogenetic diversity of plant communities contains the phylogenetic information of plants, reflects the functional information of plant communities, and can be regarded as an indicator of the functional diversity of plant communities. In this study, plant community phylogenetic diversity also showed a non-linear trend of first decreasing, increasing and then decreasing with increasing temperature (Figure 2); the CK treatment had the greatest phylogenetic diversity, and the smallest phylogenetic diversity under the greatest temperature increase, while grazing significantly reduced the phylogenetic diversity of plant communities; and the interactive experiment of gradient warming and simulated grazing also significantly affected phylogenetic diversity. Phylogenetic species variability gradually increased with the temperature increase, and grazing had no significant effect on phylogenetic species variability.

Gradient warming had a significant effect on bacterial diversity (Appendix A), the Shannon diversity of bacteria fluctuated with gradually increasing temperature, the phylogenetic diversity of bacteria showed a hump-shaped trend with temperature (Figure 3), and simulated grazing had no significant effect on bacterial diversity. The change in fungal diversity was just the opposite: although fungal diversity showed different fluctuations with the increase in gradient temperature, the increase in gradient temperature had no significant effect on fungi, while simulated grazing significantly increased fungal diversity (Appendix A).

The aboveground biomass, belowground biomass, and total biomass all showed hump-shaped trends under gradient warming (Figure 4), but the changes in aboveground biomass did not reach significant levels (Appendix A). Community stability (the degree of variation in species richness between communities) varied relatively steadily across the temperature gradient, and warming slightly increased community stability in alpine grassland, while simulated grazing did not affect community stability. To further analyze the experimental effects on biomass, we calculated the percentage of aboveground biomass and the stratified distribution of belowground biomass; the results show that the percentage of aboveground biomass accounted for approximately 40% of the total biomass, and the percentage of aboveground biomass decreased after temperature increase (Figure 5), and increased slightly with the increase in temperature. The simulated grazing did not change the distribution of biomass above or below ground. Additionally, the stratification of belowground biomass in the subsurface was not significantly affected by the gradient warming and simulated grazing (Figure 5).

With the idea of network analysis, the co-occurrence network analysis of species in the plant community showed that only a few species in the community were correlated (Appendix A), and the correlation between species was mostly positive; there were a few negative correlations in the CK + G, A + G, B + G, C + NG, C + G, G + NG and D + G treatments. Additionally, the experimental treatment did not significantly affect the correlation between species in the community (Table 1).

### 2.3. The Relationship between Biodiversity and Ecosystem Function

Factors affecting ecosystem functioning include environmental factors, which may include abiotic and biotic factors. In this study, the correlation between the environmental factors and biodiversity and ecosystem function showed that soil moisture content significantly influenced biodiversity (Figure 6), while the environmental factors related to ecosystem function were 10 cm aboveground temperature, soil temperature, soil bulk density and soil organic matter content, but only soil organic matter was significantly correlated with biomass, soil temperature and soil bulk density were only significantly correlated with belowground biomass in the third layer, and the variation in community stability was not correlated with the environmental factors.

Then, we hypothesized that the mutual relationship between aboveground organisms may be related to the function of the ecosystem, so we analyzed the correlation between the co-occurrence network analysis results of the plant community and the ecosystem function (Appendix A). The analysis results show that the network of aboveground plants only has a strong correlation with biodiversity and no correlation with ecosystem function. The environmental drivers of ecosystem structure and function were next identified using partial Mantel tests (Figure 7): the species diversity of plant communities (Shannon, Simpson and Pielou) was found to be significantly driven by 10 cm aboveground temperature and soil moisture, and the phylogenetic diversity of plant communities was significantly influenced by soil moisture, while microbial diversity was not significantly driven by environmental factors and ecosystem function was only correlated with the organic matter content of the soil.

By regressing the effects of biodiversity on ecosystem function in the experiment (Figure 8), we observed that there was no significant regression between biodiversity with community stability (degree of inter-community species variation), that there was also no linear regression between total ecosystem biomass and biodiversity, that plant communities with high stability also have widely varying total biomass, and that high community stability does not necessarily result in high productivity. Moreover, there was no significant linear relationship between biodiversity and ecosystem function under the different treatments (dots with different colors in Figure 8).

We also found that there was also no linear regression between total ecosystem biomass and biodiversity, that plant communities with high stability also have widely varying total biomass, and that high community stability does not necessarily result in high productivity. Moreover, there was no significant linear relationship between biodiversity and ecosystem function under different treatments (dots with different colors in Figure 8).

## 3. Discussion

OTC warming is a passive form of warming related to the natural environment of the region [29], and the different sizes of OTC we set up in this study yielded different magnitudes of air warming; soil surface temperature and soil moisture were also significantly affected. The vegetation type in our study area is a dwarf shrub meadow, with most of the vegetation being approximately 10 cm in height [30]. Therefore, the air temperature at 10 cm above ground in this study showed a good temperature gradient. The treatment with simulated grazing reduced standing dead material and the accumulation of dead leaves on the ground; soil–air contact was more adequate; soil temperature and moisture could be strongly correlated with temperature changes in the ground air; and the removal of standing dead material caused an increase in soil bulk, a decrease in organic matter and a deterioration in soil quality [31].

The distribution of species is the result of numerous ecological processes influenced by species evolution, geographic variation, and environmental factors, resulting in different patterns of variation in species diversity [32]. Our study found that alpine meadow plant community biodiversity did not decrease monotonically with warming, but was moderately disturbed, confirming that alpine vegetation on the Tibetan Plateau has lived in unfavorable habitats for a long time [33,34,35,36], and that future warming will have a positive impact on alpine grasslands on the Tibetan Plateau, but too strong a warming effect will still reduce alpine grassland vegetation diversity [37,38,39]. Furthermore, our study found that for vegetation on the Tibetan Plateau, a temperature increase of approximately 1 °C is the threshold limit for vegetation to survive well, and that the direct factor affecting the survival of alpine vegetation is soil moisture content. This result is consistent with the majority of studies showing that biodiversity loss is closely linked to drought [40,41,42]. Grazing reduces the number of plant species and alters the competitive relationships of species in the community, as shown in previous studies where grazing reduced the amount and height of taller grasses and shorter weeds had more ecological niches to grow [43,44,45,46]. Our study showed that simulated grazing significantly reduced phylogenetic diversity and increased phylogenetic endemism in plant communities, and slightly altered species interactions. This is the same as the results of the previous study, but we also found that simulated grazing did not significantly affect plant biodiversity, unlike the previous results; this is probably related to the way we simulated grazing, and that simulated grazing had only a very weak effect on the following year’s growth of alpine meadow plants, which was not sufficient to cause species loss.

In terms of microbial diversity, bacterial diversity showed significant changes with increasing temperature gradient. As temperature increased, bacterial diversity also increased, which is consistent with the results of previous studies [47,48], and the greatest bacterial diversity was found in the moderately increasing temperature treatment, indicating that bacteria had a threshold of tolerance to the environment; simulated grazing had no effect on bacterial diversity but significantly increased fungal diversity; and temperature had no significant effect on fungal diversity, which is related to the function of fungi—mostly saprophytic aerobic taxa [49]. Simulated grazing increased fungal diversity because grazing improved air circulation on the soil surface.

The response of grassland biomass to the warming gradient in this study showed a hump-shaped change, which is more similar to the change in biodiversity in this study, indicating a positive correlation between biodiversity and grassland biomass in this study, and suggesting that moderate warming is beneficial to grassland vegetation on the Tibetan Plateau [37]. Our study also found that warming led to a shift in vegetation biomass to the subsurface, which was associated with a decrease in soil moisture caused by warming, with more roots enhancing the ability of plants to access water [50]. Furthermore, simulated grazing in this study resulted in compensatory plant growth, with a weak increase in biomass following simulated grazing. Community stability was not significantly affected by warming and grazing. Some studies have shown that the number of species first decreases and then increases after long-term warming [51]; therefore, there was no significant change in the stability of the community calculated by the variability of species diversity.

While many previous studies have disputed the extent to which biodiversity affects ecosystem function [5,6,7], this study, after considering plant species diversity, functional aspects of diversity (phylogenetic diversity) and microbial diversity, shows that biodiversity does not contribute significantly to ecosystem productivity and community stability, and that community stability and productivity are not significantly correlated, while environmental factors partially affect ecosystem function. This could be related to the alpine environment in which the study area is located, which resulted in the vegetation being in an unsaturated state; environmental disturbances resolved this unsaturation when plant biodiversity did not change significantly, and biomass showed a dramatic response. This resulted in ecosystem function not being significantly affected by biodiversity, while environmental factors altered ecosystem function.

## 4. Materials and Methods

### 4.1. Study Site

This study was conducted at the Haibei Alpine Meadow Ecosystem Research Station (37°36′ N, 101°19′ E, 3220 m above sea level) (Appendix A), where the climate is plateau continental. The annual mean temperature is −2 °C, and the annual mean precipitation is 500 mm, of which the majority occurs during summer. The growing season extends from May to September. Mollic-Cryic Cambisol is the main soil type [52]. The main vegetation type in this area is the typical zonal vegetation of the Qinghai-Tibet Plateau. The alpine shrub meadows with *Potentilla fruticosa* as the constructive species are distributed in the shady slopes, foothills and valley lowlands of the mountains; the alpine Kobresia meadows, with *Kobresia humilis* as the constructive species, are found on the sunny slopes and beaches of the mountains; and the swampy meadows, with *Kobresia tibetica* as the dominant species, are found on the river banks. The plant community is simple in structure, with small species composition, a short growing season and low biological productivity. The main grazing animals are Tibetan sheep and yaks.

### 4.2. Experimental Design and Treatments

This field experiment was established in July 2011 within a fenced 50 × 50 m^2^ flat area, which was previously used as a winter pasture, to examine how the alpine meadow ecosystem responds to gradient warming and grazing. Fifty plots composed of five temperature treatments (ambient temperature and four warming treatments) with ten replicates were distributed into subplots divided into 10 rows and 5 columns following a random block design (Appendix A). A 2 m buffer strip between each plot was used to separate each area from its neighboring areas (Figure 9). Half blocks were treated with simulated grazing by clipping 60% of the dead standing biomass before the growing season every year, and the remaining half was not clipped (Appendix A). Four levels of warming treatments were achieved by installing four types of conical open-top chambers (OTCs, Solar Components Corporation, Manchester, NH, USA) constructed of 1.0 mm-thick fiberglass. The four types of OTCs were each 40 cm in height, but the top and bottom diameters were 1.6 m and 2.05 m (hereafter referred to as treatment (A)), 1.3 m and 1.75 m (B), 1.0 m and 1.45 m (C), and 0.7 m and 1.15 m (D), respectively. These four types of OTC treatments successfully generated increased air and soil temperature gradients.

### 4.3. Sample Collection and Soil Physicochemical Property Analysis

The Onset U23-001 (Onset Computer Corporation, Bourne, MA, USA) temperature data logger was used to record the ground temperature in the experimental area. The data logger was fixed to the center of the OTC sample (to avoid the edge effects) using a homemade iron frame and the data sensor was positioned approximately 5 cm above the ground (T.AG5). Additionally, the Onset HOBO Environmental Weather Data Logger H21-002 (Onset Computer Corporation, Bourne, MA, USA) was also set up to collect temperature at 10 cm above the ground (T.AG10) and soil temperature (T.BG5) data from two adjacent plots. All the temperatures were collected every 2 h, continuously collected throughout the year, and the data were exported when the battery was replaced.

The soil moisture content (SMC) was measured using the drying method. The soil at a depth of 10 cm was collected and placed into a sealed bag. After taking it back to the laboratory, the fresh weight was weighed, then dried at 80 °C for more than 48 h to a constant weight and then weighed. The difference between the two measurements was determined via the measurement of the soil moisture weight, the ratio of fresh weight to soil moisture content and the water content of each sample 3 times in parallel, using the mean value to represent the final moisture content (%) of each soil sample.

From late August to early September in 2019 and 2020, a sample square measuring 0.5 × 0.5 m^2^ was selected as an above-ground plant sample in each experimental sample plot to investigate plant species and numbers, and then we calculated community stability using the surveyed data, which was expressed as the inverse of the coefficient of variation of the population density of the species in the community. Given the small size of the experimental sample plot, plant samples were collected from a 0.25 × 0.25 m^2^ square, mowed and brought to the laboratory in sample bags, then dried to a constant weight and weighed as aboveground biomass (AGB). Soil bulk density (SBD) was determined using the ring knife method. Plant debris and contaminants on the soil surface of the plot were cleaned, and samples were collected at a depth of 0–10 cm using a ring knife. Two replicates were collected and placed in separate sealed bags. Samples were then taken to the laboratory and dried at 105 °C to a constant weight, and soil bulk density was calculated as the dry weight per unit volume. We used a soil auger (5 cm inner diameter) to collect 0–30 cm of soil in 3 layers throughout the experiment in the plant sampling area. Two replicates were collected from each plot, the two spiral samples were mixed to form a composite soil sample, stones visible to the naked eye were removed, and root samples were collected and placed in a sealed bag, then returned to the laboratory and dried to constant weight as the belowground biomass (BGB). The soil organic matter (SOM) content was determined using the potassium dichromate oxidation–external heating method.

### 4.4. Soil Microbial DNA Sequencing

In 2020, when collecting plants and soil, 0–10 cm of soil was collected using a soil auger with an inner diameter of 3 cm. Three soil borers were taken from each sample quadrat, 3 samples were mixed into one sample, and each sample was sieved through a 2 mm sieve after removing roots and debris. Samples were then frozen (−20 °C) until DNA was extracted. DNA was extracted using the E.Z.N.A. Soil DNA Extraction Kit (Omega Bio-Tek, Norsross, GA, USA) following the manufacturer’s instructions. For bacterial community composition, 515f/806r primer sets (515f, GTGYCAGCMGCCGCGGTAA, 806r, GGACTACNVGGGTWTCTAAT) were used to amplify (triplicate reactions for each sample) the 16S rRNA gene (cited by Earth Microbiome Project). For fungal community composition, the ITS1f/ITS2 primer pair (ITS1f, CTTGGTCATTTAGAGGAAGTAA, ITS2, GCTGCGTTCTTCATCGATGC) was selected to amplify the ITS1 region of the rRNA gene (cited by Earth Microbiome Project). PCR sequence amplification was performed using commonly used methods. The 16S rRNA amplicons and ITS amplicons were pooled separately and then sequenced with the Illumina MiSeq instrument (Illumina, San Diego, CA, USA). Raw sequence data were processed using the QIIME v1.9 pipeline, where sequences were quality filtered, chimera checked and OTU clustered, and taxonomy assignment was performed. The USEARCH algorithm was utilized to conduct chimera detection and OTUs clustering (97% similarity). Taxonomy was identified for each OTU using the RDP classifier [53] trained on the Greengenes and UNITE databases for bacterial and fungal sequences.

### 4.5. Statistical Analysis

We used muscle (version 5.1) [54] to align the DNA sequences, and then used the dist.ml and NJ functions of the ‘phangorn’ package in the R language to calculate the neighbor-joining evolutionary tree of bacteria and fungi, and finally used the ‘picante’ package pd function to calculate the phylogenetic diversity of bacteria and fungi, and the alpha diversity of bacteria and fungi was calculated using the ‘vegan’ package diversity function. The metrics of species diversity of plant communities were also calculated using the diversity function of the package ‘vegan’; the phylo.maker function of the package ‘V.PhyloMaker’ was used to construct a phylogenetic tree of the plant community; and then the functions pez.shape and pez.endemism of the ‘pez’ package in R were used to calculate the phylogenetic diversity, phylogenetic endemism [55] and phylogenetic species variability [56] of plants. To determine if there were significant differences in environmental factors, species diversity, and ecosystem functions among treatments, two-way ANOVA and Duncan’s new multiple range test were used. The relationship between environmental factors and ecosystem structure and function was further investigated using Mantel analysis based on the mantel function of the ‘vegan’ package and correlation analysis based on the ‘corrplot’ package; we also analyzed the relationship between biodiversity and ecosystem function, and the relationship between biomass and community stability through linear regression based on the rlm function of the ‘MASS’ package. All analyses were performed in the R 4.1.1 environment. The visualization of the data was performed with R software (version 4.1.1, R Foundation for Statistical Computing, Vienna, Austria) and Origin (version 2018C SR1 b9.5.1.195, OriginLab Corporation, Northampton, MA, USA), and all graphics were enhanced in Adobe Illustrator CC 2018 (version 22.0.0, Adobe Systems Incorporated, San Jose, CA, USA).

## 5. Conclusions

There is increasing evidence that grassland biodiversity loss and functional decline (grassland degradation, etc.) are closely related to climate change and anthropogenic grazing activities. The combined effects of this are still underexplored [40]. Our study analyzed the effects of simulated gradient warming and grazing on the relationship between plant diversity and productivity and community stability, which resulted in three key findings: (1) Plant biodiversity, soil microbial diversity and community productivity in alpine grasslands show fluctuating trends with temperature gradients, and a temperature increase below approximately 1 °C is beneficial to alpine grasslands; moderate grazing only increases the fungal diversity of the soil surface layer. (2) The warming shifted plant biomass underground in alpine grasslands to obtain more water in response to the decrease in soil moisture caused by the temperature rise; community stability was not affected by the warming or grazing. (3) Community stability was not significantly correlated with productivity, and environmental factors, rather than biodiversity, influenced community stability and productivity.

## Figures and Tables

**Figure 1 plants-11-01428-f001:**
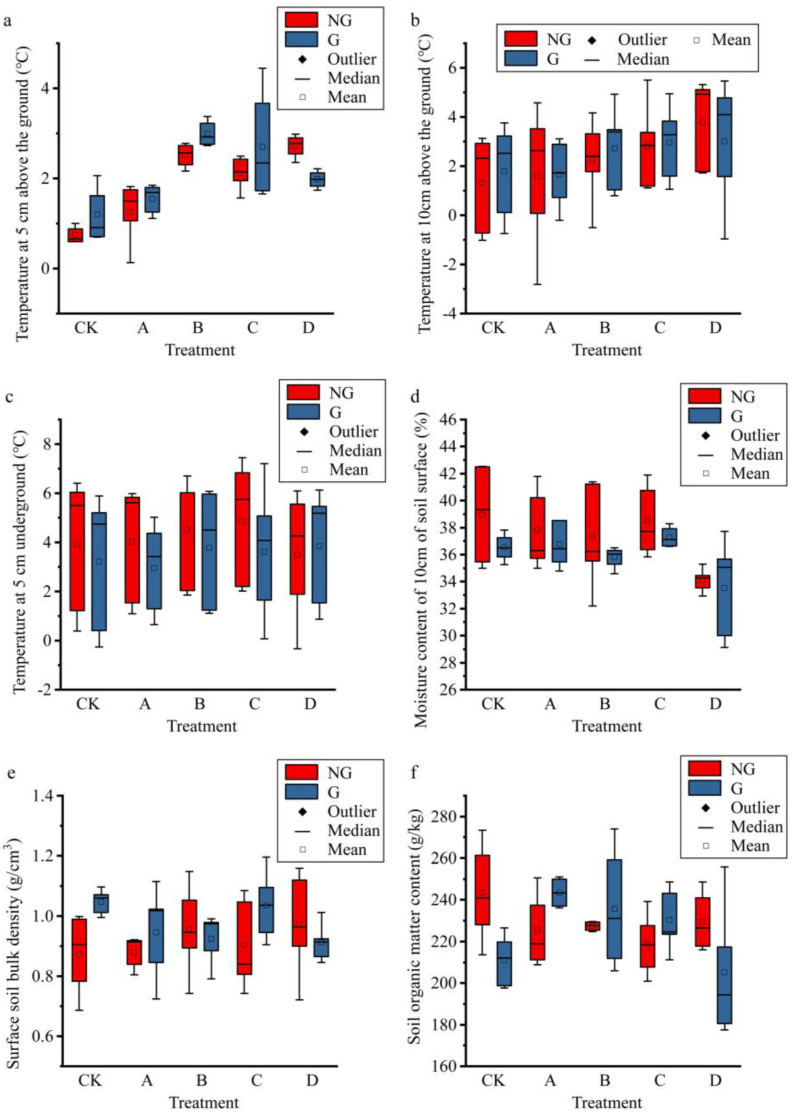
Changes in environmental factors in experimental treatments. (**a**), temperature at 5 cm aboveground; (**b**), temperature at 10 cm aboveground; (**c**), temperature at 5 cm underground; (**d**), moisture content of 10 cm soil; (**e**), soil bulk density of top 10 cm soil; (**f**), organic matter content of top 10 cm soil. The horizontal axis indicates the experimental treatments that resulted in different warming gradients. NG and G indicate treatments without simulated grazing and treatments with grazing, respectively.

**Figure 2 plants-11-01428-f002:**
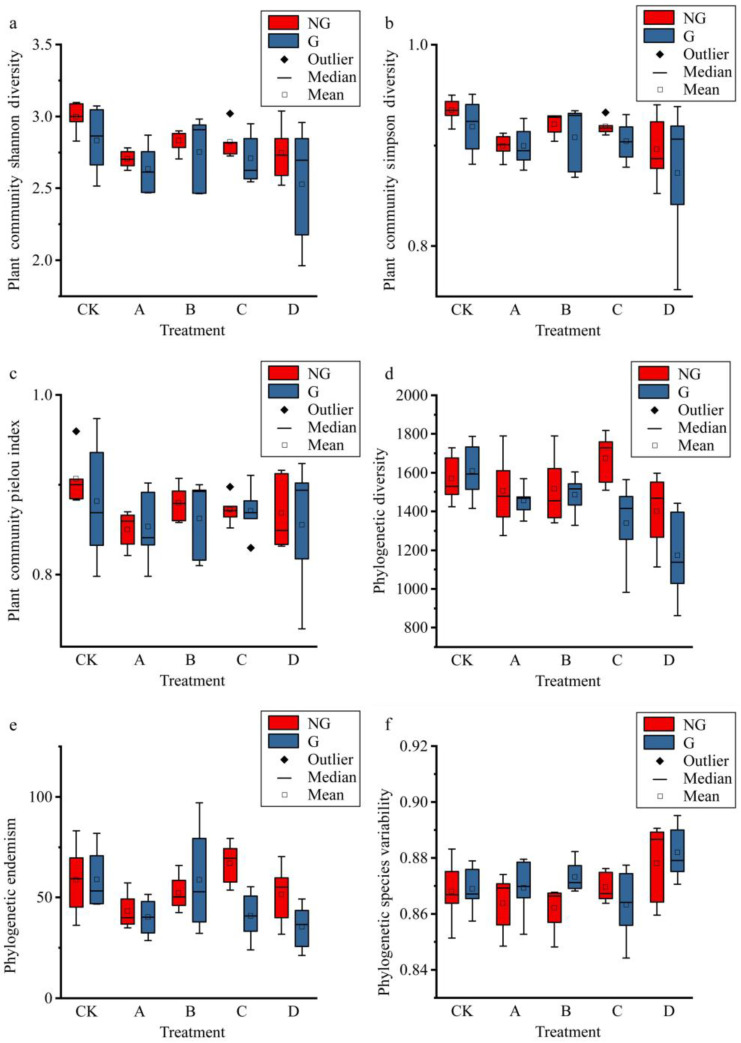
Effects of gradient warming and grazing on biodiversity of plant community. (**a**), plant community shannon diversity index; (**b**), plant community simpson diversity index; (**c**), plant community pielou diversity index; (**d**), plant community phylogenetic diversity index; (**e**), plant community phylogenetic endemism index; (**f**), plant community phylogenetic species variability index. The horizontal axis indicates the experimental treatments that resulted in different warming gradients. NG and G indicate treatments without simulated grazing and treatments with grazing, respectively.

**Figure 3 plants-11-01428-f003:**
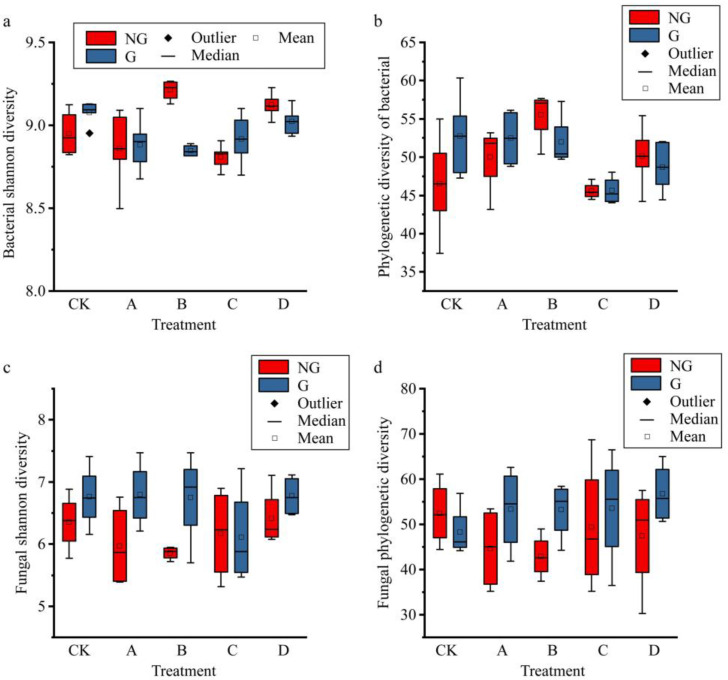
Effects of gradient warming and grazing on biodiversity of microbial community. (**a**), shannon diversity index of bacterial community; (**b**), phylogenetic diversity index of bacterial community; (**c**), shannon diversity index of fungal community; (**d**), phylogenetic diversity index of fungal community. The horizontal axis indicates the experimental treatments that resulted in different warming gradients. NG and G indicate treatments without simulated grazing and treatments with grazing, respectively.

**Figure 4 plants-11-01428-f004:**
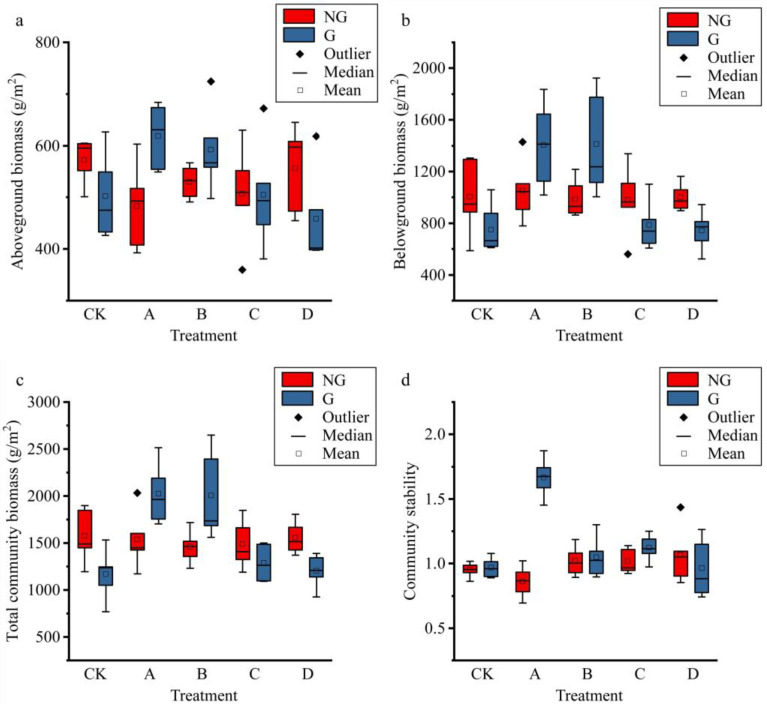
Effects of gradient warming and grazing on ecosystem functioning. (**a**), aboveground biomass of plants; (**b**), belowground biomass of plants; (**c**), total biomass of plants; (**d**), structural stability of plant community. The horizontal axis indicates the experimental treatments that resulted in different warming gradients. NG and G indicate treatments without simulated grazing and treatments with grazing, respectively.

**Figure 5 plants-11-01428-f005:**
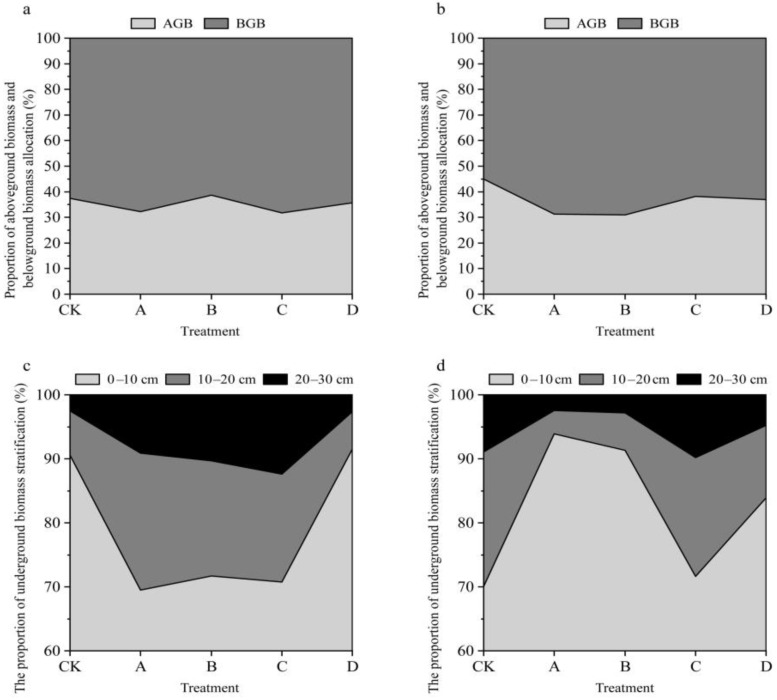
Effect of gradient warming and grazing on biomass distribution. (**a**), changes in the proportion of above-ground biomass and below-ground biomass allocated with the warming gradient when not grazed; (**b**), changes in the proportion of above-ground biomass and below-ground biomass allocated with the warming gradient when grazed; (**c**), changes in the proportion of below-ground biomass stratified with the warming gradient when not grazed; (**d**), changes in the proportion of below-ground biomass stratified with the warming gradient when grazed. The horizontal axis indicates the experimental treatments that resulted in different warming gradients.

**Figure 6 plants-11-01428-f006:**
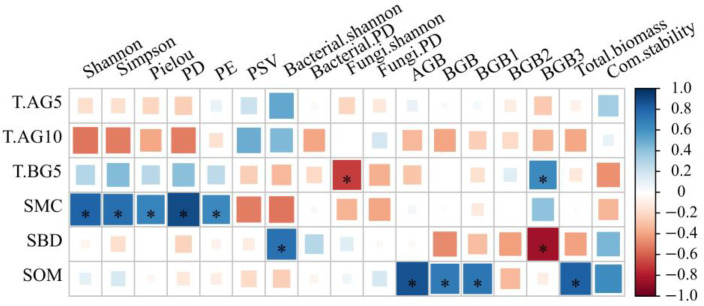
Correlation of environmental factors with biodiversity and ecosystem function. * indicates the significance of correlation (*p* = 0.05); T.AG5, T.AG10, T.BG5, SMC, SBD and SOM indicate temperature at 5 cm above ground, temperature at 10 cm above ground, temperature at 5 cm in soil, soil moisture content, soil bulk density and soil organic matter, respectively; PD, PE and PSV indicate phylogenetic diversity, phylogenetic endemism and phylogenetic species variability, respectively; AGB and BGB denote aboveground biomass and belowground biomass; BGB1, BGB2 and BGB3 denote belowground biomass at 0–10 cm, 10–20 cm and 20–30 cm, respectively.

**Figure 7 plants-11-01428-f007:**
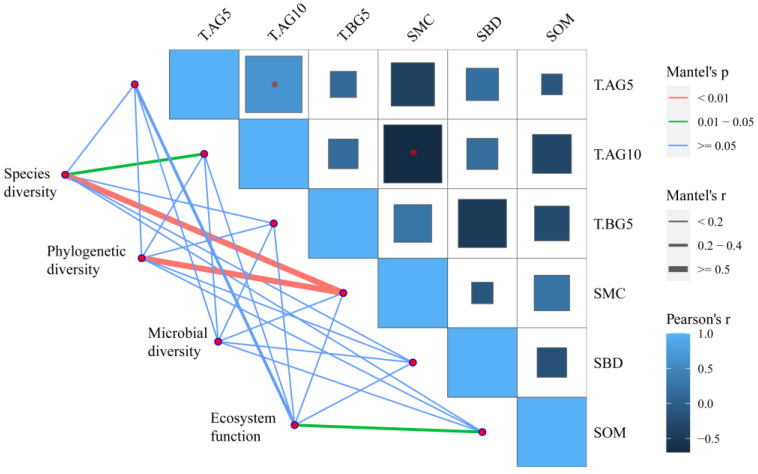
Environmental drivers of ecosystem structure and function. Pairwise comparisons of environmental factors are shown, with a color gradient denoting Spearman’s correlation coefficients. Ecosystem structure (biodiversity) and ecosystem function (biomass, biomass distribution and community stability) were related to each environmental factor by partial Mantel tests. Edge width corresponds to the Mantel’s r statistic for the corresponding distance correlations, and edge color denotes the statistical significance based on 9999 permutations. T.AG5, T.AG10, T.BG5, SMC, SBD and SOM indicate temperature at 5 cm above ground, temperature at 10 cm above ground, temperature at 5 cm in soil, soil moisture content, soil bulk density and soil organic matter, respectively. * indicates the significance of correlation (*p* = 0.05).

**Figure 8 plants-11-01428-f008:**
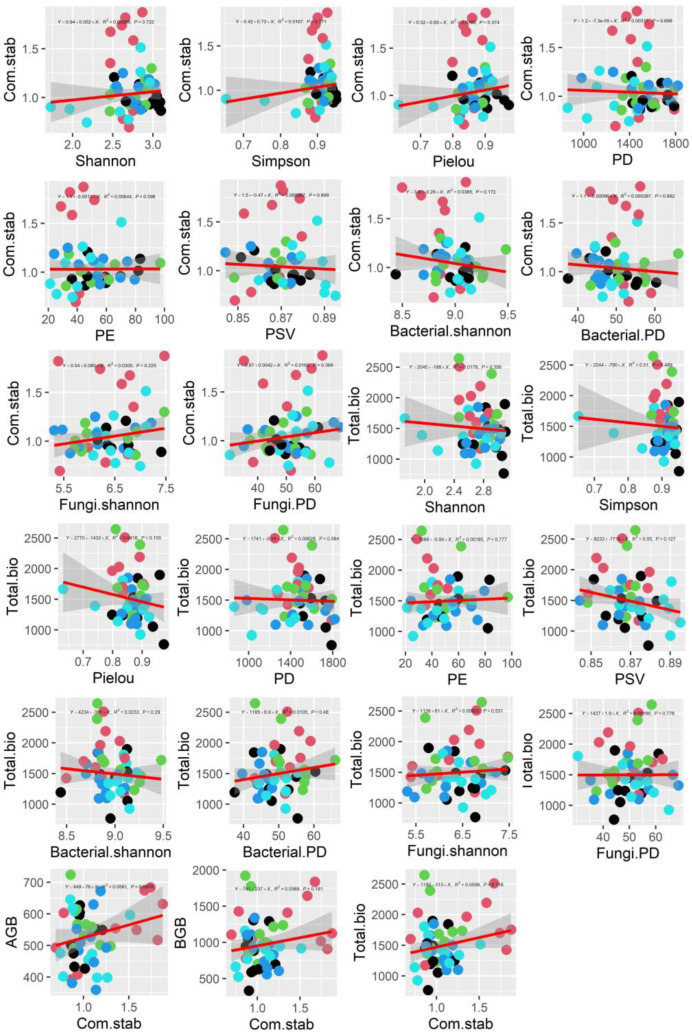
The relationship between biodiversity and biomass and community stability, and the relationship between biomass and community stability. Different colored dots indicate different treatments, gray translucent areas indicate 95% confidence intervals. Shannon, Simpson and Pielou indicate different indices of diversity. PD, PE and PSV indicate phylogenetic diversity, phylogenetic endemism and phylogenetic species variability, respectively. AGB and BGB denote aboveground biomass and belowground biomass. Com.stab indicates community stability.

**Figure 9 plants-11-01428-f009:**
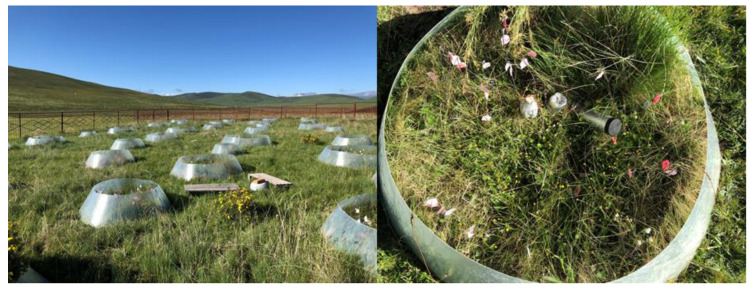
Map of treatment plots and OTC samples.

**Table 1 plants-11-01428-t001:** Results of network co-occurrence analysis of plant communities under different treatments.

Treatment *	Average Degree	Average Weighted Degree	Modularity	Statistical Inference	Graph Density	Nodes	Edges
CK + NG	0.577	0.577	0.507	226.254	0.011	52	15
CK + G	0.706	0.627	0.736	230.432	0.014	51	18
A + NG	0.375	0.375	0.519	190.919	0.008	48	9
A + G	0.851	0.766	0.847	215.927	0.019	47	20
B + NG	0.356	0.356	0.688	175.426	0.008	45	8
B + G	0.304	0.130	0.333	176.360	0.007	46	7
C + NG	0.600	0.440	0.975	218.481	0.012	50	15
C + G	0.809	0.723	0.268	213.521	0.018	47	19
D + NG	0.682	0.591	0.814	190.962	0.016	44	15
D + G	0.732	0.634	0.423	177.897	0.018	41	15

* CK, A, B, C and D, indicate control and the experimental treatments that resulted in different warming gradients; NG and G, indicate no simulated grazing and with simulated grazing, respectively; + indicates superposition of two treatments.

## Data Availability

The data presented in this study are available on request from the corresponding author. The data are not publicly available due to privacy.

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
