# Peer review of "Biodiversity and Ecosystem Function under Simulated Gradient Warming and Grazing"

_plants, 2022, doi:10.3390/plants11111428_

Round 1

Reviewer 1 Report

No.

Author Response

We would like to thank you for your careful reading, helpful comments, and constructive suggestions, which has significantly improved the presentation of our manuscript.

We are very sorry for the mistakes in this manuscript and inconvenience they caused in your reading. The manuscript has been thoroughly revised and edited by a native speaker, so we hope it can meet the journal's standard. We also have rewritten the confusing sentences in the article and hope this will help you understand the article.

Once again, we thank you for the time you put in reviewing our paper and look forward to meeting your expectations. Since your inputs have been precious, in the eventuality of a publication, we would like to acknowledge your contribution explicitly.

Reviewer 2 Report

I have reviewed with interest your work. The manuscript entitled „Biodiversity and ecosystem function under simulated gradient warming and grazing" (Manuscript ID: plants-1712931). This study concentrated to evaluate the changes in biodiversity and ecosystem function of grasslands in Qinghai-Tibet Plateau through an interactive experiment with simulated gradient warming and simulated grazing in an alpine grassland conducted for eight years. The article deals with a very important issue, is interesting and written very well. Some remarks are listed below:

The issue of why winter grazing was taken for analysis should be better justified and explained in the introduction. Is it truly possible to graze in such a harsh climate of Tibetan Plateau in winter?

Figures 1-8 – any abbreviations used in the figure should be explained below the figure.

Table 1 – the same as above.

Discussion line 303 - the claim that biodiversity does not affect ecosystem function in most world’s researches is very risky and must be supported by solid evidence that cannot be provided by simulation-based work alone. Authors should discuss this proposal in depth.

Reviewer 3 Report

In this study the authors aim at testing the relationship between diversity and ecosystem function in response to an artificial temperature and grazing gradient, focusing on plant and microbial communities. In general, the authors did not find strong links between biodiversity and ecosystem function. Instead, ecosystem function seemed to be more affected by environmental conditions. For example, warming which increased plant diversity in alpine meadows but decreased ecosystem function. Instead, simulated grazing can reduce productivity decline and can be used to manage the negative effects of warming… (not clear neither in the abstract nor in the discussion/conclusions).  

In my opinion, the idea of this study is interesting and useful, especially considering the increasing impact of warming and grazing. The approach of analyzing the correlation between ecosystem function and biodiversity is innovative and updated.

Some general comments I have are:

After reading the abstract it is not clear to the reader why study microbes and plants at the same time. Also, the concluding remark in the abstract, stating that grazing can be used to manage the negative effects of warming on alpine meadows is difficult to understand. This concluding sentence seemed odd to me.

The results are difficult to follow, because of the overuse of acronyms and lack of clarity in the figures and tables. Some information is missing (see below).

The discussion contains too much repetitive information on the results.

Based on the above and the detailed comments that follow, and given the relevance of the study, I would say that the article can be published after major revision

Specific comments follow:

Abstract

It needs to be improved. Please explain if the experiments were performed in the laboratory or in the field. The concluding sentence (last sentence) is not clear, and seems odd… After reading the abstract I do not understand how grazing can help plants overcome the impact of a warming climate.

Graphical abstract

Absent

Highlights

I did not find highlights.

Introduction

It is interesting and well-written.

L 48: other ecosystem functions of ecosystems… delete “of ecosystems”, it is redundant.

Throughout the introduction the authors repeatedly mention “ecosystem function”, but they are not specific of which functions they are talking about. See for example, L 72, 76, 86, 88, 90, 91

L 84 Can you provide an example of anthropic disturbance?

L 84-85 which attributes of ecosystem structure and function are you talking about? Can you be more specific?

Methods

Could you please add a picture of the OTC used for the artificial warming experiment? It would be very useful.

Based on the goals of the study, it is not clear why analyze contaminants on the soil surface. Also, why DNA sequencing? To assess biodiversity of microbes?

Results

In general, the results were difficult to follow because of the overuse of acronyms which are not explained in the figures and tables. Can you avoid the acronyms? In addition, the figures are hard to understand because they lack essential descriptions of axis X.

Figure 1. For clarity, please explain in the figure legend what CK, A, B, C and D mean. Figures should be easily understandable independently of the text. Thus, this information is necessary. Even after reading the text, it is not clear what the letters in axis X represent. Thus, it is difficult to understand these results. The same comment applies to figures 2-5.

L- 105 What do you mean by: “Grazing increased environmental factors? What not say: above ground temperature etc? It is confusing.

Table 1. Please explain the acronyms…. Specifically, what do A, B, C and D stand for?

How did you measure community stability? Lines 202, 209, 210. It is not clear.

Discussion

Please, do not repeat the results in the discussion (Lines 226-254; 261-266; 287-295).

Again, community stability is mentioned although it is not clear on how it was measured nor which parameters were used to measure stability.

I would not state so emphatically that biodiversity does not affect ecosystem functioning, given that only a handful of functions were considered (Lines 300-310).

Conclusion

Please do not summarize the study. Instead, write briefly the take home message.

The conclusions discuss ideas that are not based on any result shown in the article. For example, why does winter grazing mitigate the negative effects of rising temperatures? From my understanding, grazing affected taller plants (and thus biodiversity… do you mean that smaller plants are more tolerant to warming? Do we want to lose taller species? This reasoning is not clear.

Finally, I would not end my article stating that further studies are necessary… this is always the case.

Round 2

Reviewer 3 Report

The authors worked a lot and largely improved their article, which I really enjoyed.

Some very minor comments next.

The abstract is largely improved and is clearer now. Only I found lines 17-21 a bit repetitive.

Nice picture of the OTC. Thanks!

Figures 1, 2, 3, 4, 5. I think I was not clear in my previous revision. I think the authors should explain the meaning of CK, A, B, C and D from the X axis. What temperatures are represented? Same comment for Table 1
